# Local Dynamic Stability of Trunk During Gait Can Detect Dynamic Imbalance in Subjects with Episodic Migraine

**DOI:** 10.3390/s24237627

**Published:** 2024-11-28

**Authors:** Stefano Filippo Castiglia, Gabriele Sebastianelli, Chiara Abagnale, Francesco Casillo, Dante Trabassi, Cherubino Di Lorenzo, Lucia Ziccardi, Vincenzo Parisi, Antonio Di Renzo, Roberto De Icco, Cristina Tassorelli, Mariano Serrao, Gianluca Coppola

**Affiliations:** 1Department of Medico-Surgical Sciences and Biotechnologies, Sapienza University of Rome Polo Pontino ICOT, 04100 Latina, Italy; stefanofilippo.castiglia@uniroma1.it (S.F.C.); gabriele.sebastianelli@uniroma1.it (G.S.); chiara.abagnale@uniroma1.it (C.A.); francesco.casillo@uniroma1.it (F.C.); dante.trabassi@uniroma1.it (D.T.); cherub@inwind.it (C.D.L.); mariano.serrao@uniroma1.it (M.S.); 2Department of Brain and Behavioral Sciences, University of Pavia, 27100 Pavia, Italy; roberto.deicco@unipv.it (R.D.I.); cristina.tassorelli@unipv.it (C.T.); 3IRCCS-Fondazione Bietti, 00184 Rome, Italy; lucia.ziccardi@fondazionebietti.it (L.Z.); vincenzo.parisi@fondazionebietti.it (V.P.); antoniomp777@hotmail.it (A.D.R.); 4Headache Science and Neurorehabilitation Unit, IRCCS Mondino Foundation, 27100 Pavia, Italy; 5Movement Analysis Laboratory, Policlinico Italia, 00162 Rome, Italy

**Keywords:** motion sensitivity, Lyapunov’s exponent, pelvic rotation, postural balance, accelerometry, movement analysis, headache

## Abstract

Background/Hypothesis: Motion sensitivity symptoms, such as dizziness or unsteadiness, are frequently reported as non-headache symptoms of migraine. Postural imbalance has been observed in subjects with vestibular migraine, chronic migraine, and aura. We aimed to assess the ability of largest Lyapunov’s exponent for a short time series (sLLE), which reflects the ability to cope with internal perturbations during gait, to detect differences in local dynamic stability between individuals with migraine without aura (MO) with an episodic pattern between attacks and healthy subjects (HS). Methods: Trunk accelerations of 47 MO and 38 HS were recorded during gait using an inertial measurement unit. The discriminative ability of sLLE was assessed through receiver-operating characteristics curves and cutoff analysis. Partial correlation analysis was conducted between the clinical and gait variables, excluding the effects of gait speed. Results: MO showed higher sLLE values, and reduced pelvic rotation, pelvic tilt, and stride length values. sLLEML and pelvic rotation showed good ability to discriminate between MO and HS and were correlated with the perceived pain, migraine disability assessment score, and each other. Conclusions: these findings may provide new insights into the postural balance control mechanism in subjects with MO and introduce the sLLEML as a potential measure of dynamic instability in MO.

## 1. Introduction

Motion sensitivity symptoms, such as dizziness or unsteadiness, are frequently reported as non-pain symptoms of migraine [1,2]. Several possible explanations have been put forward to explain this symptomatology, such as a malfunctioning of the multisensory integration process [3] and peripheral and central disarrangements [4,5], including structures of the inner ear, brainstem [6], basal ganglia, and cerebellum [7], leading to a mismatch between proprioceptive cues and vestibular [8] or visual stimuli [9]. Balance impairment has also been described in subjects with migraine, with imbalance increasing with the manipulation of sensory inputs while standing [10]. Several studies were performed on migraines using the sensory organization test, which is performed through a dynamic posturography system designed to assess quantitatively an individual’s ability to use visual, proprioceptive, and vestibular cues to maintain postural stability while standing. With this methodology, differences in postural balance between the subdiagnoses of migraine have been described, with greater instability and fall risk in subjects with vestibular migraine, chronic migraine, and migraine with aura [10,11,12], whereas subjects with episodic migraine showed less pronounced balance alterations, compared with healthy controls and subjects with chronic migraine [13]. However, regardless of the presence of chronic migraine and aura, subjects with migraine may experience abnormalities during mobility tasks and walking, such as reduction in gait speed and increased step width [14,15], which are common signs of dynamic imbalance [16,17]. This suggests the presence of dynamic instability also in subjects with episodic migraine without referring the experience of aura. In this way, instrumented assessing of balance during gait inertial measurement units (IMUs) may provide insights into the subtle postural instability among the subcategory of subjects with migraine that cannot be found out through the somatosensory orientation test [18]. Inertial measurement units (IMUs), which extract gait parameters from acceleration and angular velocity data, are widely used for instrumented gait analysis due to their ease of use and ability to retrieve motion data in real-world, non-laboratory settings. Aside from the spatiotemporal and pelvic kinematic gait parameters, trunk acceleration-derived gait quality indexes can be calculated using lower-trunk acceleration signals. Particularly, nonlinear analysis of the trunk acceleration patterns through the short-term maximal Lyapunov’s exponent (sLLE) has proven to accurately characterize gait imbalance in several neurologic conditions [19,20,21,22]. sLLE quantifies the rate of divergence of the trajectories in a system’s state space over a short period and assesses local dynamic stability by measuring the sensitivity of the system to small perturbations in the initial conditions, where a positive exponent indicates divergence (instability), and a negative exponent indicates convergence (stability) of trajectories in the short term [23,24].

The aim of this study was to gain new insights into the mechanisms of motion sensitivity symptoms in individuals affected by migraine without aura (MO) studied during the inter-ictal phase. Therefore, using a wearable device, we assessed the ability of the sLLE to detect differences in local dynamic stability between MO and healthy subjects. Greater knowledge of how migraine symptomology and related disability interfere with locomotor demands in patients with MO may facilitate targeted intervention strategies. We hypothesized that the MO may exhibit dynamic imbalance due to abnormalities of the multisensory integration process, and that the sLLE, a measure of the ability to cope with internal perturbations during gait, might reflect local dynamic instability in MO.

## 2. Materials and Methods

### 2.1. Participants

This cross-sectional study was conducted at the Traumatic Orthopedic Surgical Institute (ICOT) in Latina, Italy, between March 2022 and April 2024. Fifty-six MO were screened for eligibility and forty-seven MO were included in this study. Inclusion criteria were (i) the diagnosis of migraine without aura according to the International Classification of Headache Disorders (III edition) [25]; (ii) episodic migraine pattern, defined as 1 to 14 monthly migraine days during the preceding 3 months; (iii) migraine during the interictal period at the day of the assessment, i.e., at least 3 days since the last and the next migraine attack; and (iv) being without any migraine prevention during the preceding 3 months. Exclusion criteria were (i) the presence of other primary or secondary headaches, including migraine with aura; (ii) concomitance with the menstrual period for female subjects; (iii) orthopedic, neuro-ophthalmologic, and neurologic conditions other than migraine; and (iv) ongoing pharmacological therapy, including migraine preventives other than acute migraine attack medications, and contraceptives. The presence of neuro-ophthalmologic disease was verified through examination that included a visual acuity test, an intraocular pressure measurement, and indirect ophthalmoscopy. Subjects were administered the migraine disability assessment (MIDAS), the most commonly used questionnaire to assess 3-month migraine-related disability, and the headache impact test (HIT-6) scales to assess the impact of migraine on functioning of subjects over one month, the 12-item allodynia symptom checklist (ASC-12), and the numeric pain rating scale (NPRS) to record the perceived pain during migraine attacks [26,27,28].

For group comparison, a group of healthy subjects (HS) matched for age and gait speed was enrolled. A 1:1 optimal data matching procedure using the propensity score difference method was performed to match patients with MO with healthy subjects (HS) [29]. From a dataset of 96 HS, as a result, 38 HS were included after the matching procedure, whose effectiveness was assessed using an independent sample t-test using age and gait speed as the variables. Every participant in the study was required to complete a headache diary, which was sent to them by mail at least three months prior to their initial visit. The characteristics of the included sample are described in Table 1. All participants provided informed consent prior to the experimental procedure. The study was approved by the local ethics committee (CE Lazio 2, protocol number 0139696/2021).

### 2.2. Procedures

Participants were instructed to walk along a straight path that measured 30 m in length, at a speed that they personally chose as their favorite walking pace. The corridor floor was linoleum, with no visible pavement joints or demarcation lines, and indirect lighting was evenly distributed along the path. Before the experiment, participants were instructed to walk along the trail to familiarize themselves with the procedure. There were no adverse events recorded during the procedures. There were no external stimuli given during the task. The trunk acceleration signals were recorded at a frequency of 100 Hz using a single magneto-inertial measurement unit (GSensor, BTS, Milano, Italy). The unit was positioned at the L5-S1 level and secured to the pelvis with a Velcro belt. Data were acquired through the GStudio software (version 3.5.25.0, GStudio, BTS, Milano, Italy) using the “Walk +” embedded tool, and spatio-temporal gait parameters, and pelvic kinematics were calculated. To ensure a steady-state walking assessment, we removed the first and last two strides from each 30 m walk. Any gait trials that had fewer than 20 accurately recorded consecutive strides were not included in the study [30,31,32].

### 2.3. sLLE Calculations

The short-term maximum finite-time Lyapunov’s exponent (sLLE) is a nonlinear metric that represents the stability of a dynamic system as the average logarithmic rate of divergence between the system’s trajectory and its closest neighboring trajectory. Convergent trajectories indicate local dynamic stability, while divergent trajectories indicate local dynamic instability. As a result, when applied to gait data, it represents a measure of the ability to cope with internal perturbations during gait, thus reflecting local dynamic instability [33,34]. It was calculated based on the acceleration patterns for antero-posterior (sLLEAP), medio-lateral (sLLEML), and vertical (sLLEV) directions according to the Rosenstein’s algorithm using the Lyaprosen toolbox for nonlinear time series analysis in the MATLAB environment (MATLAB 7.4.0, MathWorks, Natick, MA, USA). Twenty consecutive strides were considered for the calculations, and the acceleration signals were time-normalized to obtain 100 datapoints per stride [35]. The embedding dimension was calculated using the false nearest neighbor method, and time delay was calculated as the first minimum of the average mutual information (AMI) function [35]. Consequently, in this study, an embedding dimension = 5 and time delay = 10 were used for multidimensional state space reconstruction from the recorded one-dimensional time series data by juxtaposing the original data and delayed copies (Figure 1). Higher sLLE values reflect higher local dynamic instability.

### 2.4. Statistical Analysis

A sample size of at least 48 subjects, 24 MO, and 24 controls, was calculated to identify a good ability (AUC  >  0.70) to discriminate between MO and controls at a 95% significance level and 80% power.

Statistical analysis was conducted using JASP software (Version 0.17.2.1), and NCSS 2023 software. Correlation analysis was conducted using the “Pingouin” Python package [36], vers. 0.5.3.

After verifying the homogeneity of the variances and the normality of the distributions using the Levene’s test and Shapiro–Wilk test, and an independent sample t-test or Mann–Whitney test was implemented to assess the differences in gait parameters between MO subjects and HS. Cohen’s effect size (d) was also calculated to assess the magnitude of the differences.

Receiver operating characteristics (ROC) were plotted for the significant gait variables, and the area under the ROC curve (AUC) was calculated to assess the overall ability of the significant gait variables to discriminate between the groups. AUC values ≥ 0.70 were considered as reflecting good discriminative ability. The optimal cutoff point (OCP) was calculated as the value that maximizes the sum of sensitivity and specificity, and positive and negative likelihood ratios (LR+ and LR−, respectively) at the OCP were calculated. Positive and negative post-test probabilities (PTP+ and PTP−) were calculated by transforming the likelihood ratios through a Fagan’s nomogram to estimate the likelihood of correctly classifying at the OCP. To improve the generalizability of the results, the 12% [37] prevalence of episodic migraine was used as the prior probability in the post-test probabilities calculations.

Spearman’s partial correlation coefficients (*ρ*) excluding the effects of gait speed were conducted to assess the correlation between the gait variables with good discriminative ability and the clinical and gait variables. To account for tied scores, the tie correction factor was applied to the correlation coefficients using the following formula:$$\rho_{s}=1-\frac{6\sum d_{i}^{2}}{n\left(n^{2}-1\right)-\sum \left(t_{i}^{3}-t_{i}\right)}$$
where *d_i_* is the difference between the ranks of the values corresponding to the two variables, *n* is the number of observations, and *t_i_* is the number of tied ranks for each tie group. The term $\sum \left(t_{i}^{3}-t_{i}\right)$ is the sum of the cubes of the number of ties minus the number of ties for each distinct number of tied values, summed over all sets of ties.

## 3. Results

Forty-seven MO subjects, aged 34.13 ± 13.89 years, of whom thirty-eight (81.25%) were females, and walking at an average speed of 1.16 ± 0.17 m/s, were included. Subjects had been diagnosed with MO since 19.82 ± 12.21 months, with an average of 5.64 ± 4.63 days of migraine per month, lasting 43.85 ± 40.38 h on average. NSAIDs were the most common type of symptomatic medication, with an average of 6.79 ± 6.30 doses per month. Subjects with MO were assessed 10.64 ± 15.15 days since the last migraine attack. The included 38 healthy participants were 38.27 ± 12.46 years old, 27 of whom were females, and walking at an average gait speed of 1.21 ± 0.15 m/s. As a result of the matching procedure, no significant differences in age and gait speed were found between subjects with MO and HS (age: *p* = 0.15, Cohen’s d = 0.31; gait speed: *p* = 0.11, and Cohen’s d = 0.36).

### Trunk Acceleration-Derived Gait Indexes

Compared with HS, subjects with MO resulted in higher sLLEAP (*p* < 0.01; d = 0.66), sLLEML (*p* < 0.01; d = 0.89), and sLLEV (*p* < 0.01; d = 0.71) and reduced pelvic rotation (*p* <0.01; d = 0.91), pelvic tilt (*p* < 0.01; d = 0.83), and stride length values (*p* = 0.03; d = 0.45) (Figure 2).

sLLEML and pelvic rotation showed good ability to discriminate between MO and HS (Table 2). After adjusting the post-test probabilities based on the 12% prevalence of episodic migraine in the general population, sLLEML values higher than 1.18 and pelvic rotation values lower than 11.50° showed 73% and 70% probability to correctly classify patients with MO (Table 2).

Regardless of gait speed, sLLEML positively correlated with MIDAS score (*ρ* = 0.43, *p* = 0.01), and negatively correlated with pelvic rotation (*ρ* = −0.47, *p* = 0.00) and NRPS (*ρ* = −0.33, *p* = 0.03). Pelvic rotation values also negatively correlated with NPRS (*ρ* = −0.35, *p* = 0.00) and MIDAS score (*ρ* = −0.35, *p* = 0.03).

## 4. Discussion

The objective of this study was to investigate gait indexes reflecting mechanisms of motion sensitivity symptoms in a well-characterized group of people suffering from episodic migraine without aura between attacks by the evaluation of the ability of the sLLE to detect gait instability. We found that, regardless of age and gait speed, patients with MO exhibit greater sLLE values in all spatial directions compared with healthy subjects (Figure 2). Particularly, sLLE values in the ML direction ≥ 1.18 characterized MO with 73% probability (Table 2). Moreover, correlation analysis revealed that the sLLEML reflects the level of pain and disability caused by migraine, with greater gait local instability correlating with greater disability and severity of headache, as measured by the MIDAS questionnaire and NRPS scale, respectively. No correlations between sLLE and disease duration, the duration of the attacks, the monthly migraine days, nor the days passing since the last migraine attack at the moment of the gait assessment were found, suggesting that subjects with MO present local dynamic instability regardless of the global disease activity.

Postural balance impairment has been described in subjects with vestibular migraine and chronic migraine, with subjects with episodic migraine reporting similar postural balance findings to healthy controls [11,38,39]. However, these observations are based on static balance tests or on the somatosensory orientation test, which does not assess the dynamic behavior during natural gait [40,41,42]. There is much compelling evidence that subjects with episodic migraine without aura do exhibit significant cortical dysexcitability, abnormalities in executive functions, and integrative pain processing [43,44], which could influence sensorimotor activity. Consequently, effective balance function and motion, i.e., dynamic stability, may be impaired as a result. Furthermore, data from functional neuroimaging studies show altered brain network patterns committed to multisensory integration, including sensorimotor and executive control, in migraine without aura [45,46]. These functional changes affect key brain regions related to pain perception, autonomic responses, gait control, and cognitive processing, suggesting that migraine might hinder the ability of the brain to process and integrate sensory information effectively [46]. We argue that this interictal disrupted sensory processing, in conjunction with the well-known abnormalities in the visual processing [3,47], may contribute to dynamic instability by reducing individuals’ ability to respond to environmental changes and maintain balance. Therefore, the findings of our study suggest that assessing local dynamic stability during gait through the sLLEML may detect the subtle abnormalities in dynamic balance control also in subjects with episodic migraine without aura as a result of the impairment of multisensory integration process [3].

Furthermore, we found that participants with MO exhibited lower pelvic rotation range during gait (Figure 2) compared with HS, and that pelvic rotation values lower than 11.50° characterized MO with 70% probability (Table 2). Reduction in pelvic rotation was also correlated with higher sLLEML values, MIDAS questionnaire, and NPRS. Subjects with migraine have been described to present decreased mobility of the cervical spine and increased stiffness of neck muscles [48]. Moreover, the results are consistent enough with other studies reporting the sLLEML to correlate with pelvic mobility [49]. Therefore, another possible explanation of our results is that patients with MO stiffen their axial movements in the context of cautious gait pattern due to perceived pain and instability due to multisensory integration impairment, thus increasing local instability of the trunk acceleration patterns. The other spatiotemporal gait parameters showed no significant differences when compared to healthy subjects, except for a significant reduction in stride length (Figure 2), which did not show to characterize the gait abnormalities of participants with MO with adequate probability (Table 2). In this way, given sLLEML’s ability to reflect the inability to cope with small perturbations, it may be considered a useful tool for identifying subjects with MO who experience dynamic instability during gait.

This study presents several limitations. Although inclusion criteria satisfied the definition of migraine without aura according to the International Classification of Headache Disorders, a complete vestibular screening was not conducted. Furthermore, although we used the actual prevalence of episodic migraine in the general population as the pre-test probability in the post-test probability calculations, our sample was relatively small and did not reflect the prevalence of migraine. Therefore, further studies with broader populations are needed to confirm our findings.

## 5. Conclusions

Subjects with episodic migraine without aura exhibit abnormalities in trunk local dynamic stability during gait, as measured by the sLLE in the medio-lateral direction. This parameter is correlated with disease-related disability levels, severity of pain during the attacks, and reduction in pelvic rotation during gait. The sLLE in the ML direction can be used as an index to quantify gait instability and the ability to face small perturbations during gait in subjects with episodic migraine. Considering the previous evidence of increased postural instability during the migraine attack compared with the interictal phase [42], it is of interest to test whether the abnormalities noted here during the interictal period become more manifest during the attack and in proportion to its severity.

## Figures and Tables

**Figure 1 sensors-24-07627-f001:**
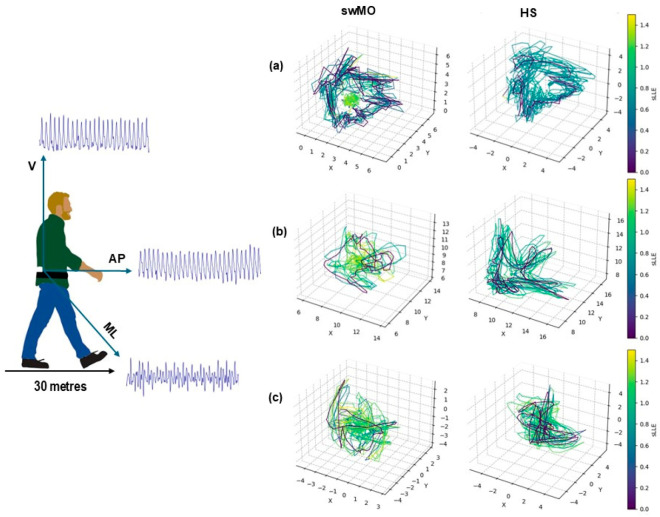
Phase space reconstruction plots of a representative patients with MO (plots on the (**left**)) and a healthy subject (plots on the (**right**)) for the (**a**) vertical (V), (**b**) antero-posterior (AP), and (**c**) medio-lateral (ML) directions of the acceleration signals. The color of each line segment corresponds to the sLLE value at that point in time, as indicated by the colormap. Lighter colors indicate higher divergence levels.

**Figure 2 sensors-24-07627-f002:**
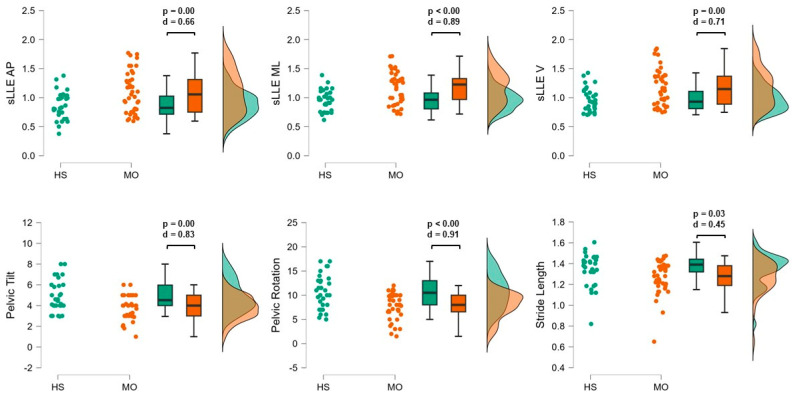
Significant differences between patients with MO and HS. The figure represents the raincloud plots of the significantly different variables between MO (orange) and HS (green). *p*-values and Cohen’s d are reported above the boxplots. Distribution shapes are also reported for each plot for each group. Created using JASP software, vers. 0.17.02.

**Table 1 sensors-24-07627-t001:** Clinical and demographic features, spatio-temporal gait parameters, and pelvic kinematics of study population. MO, subjects with migraine without aura; MIDAS, migraine disability assessment test; HIT-6, headache impact test; ASC-12, allodynia symptom checklist; sLLE, short-term Lyapunov’s exponent; AP, ML, V, antero-posterior, medio-lateral, and vertical direction of the acceleration signals, and NSAIDs, non-steroidal anti-inflammatory drugs.

		MO (N = 47)		HS (N = 38)	
		Mean	SD	Mean	SD
Age		34.13	13.89	38.27	12.46
Sex	Females	38 (81.25%)		27 (71.79%)	
	Males	9 (18.75%)		11 (28.20%)	
Disease duration (months)		19.82	12.21		
N. migraine days/month		5.64	4.63		
Duration of migraine attacks (hours)		43.85	40.38		
Days since the last attack		10.64	15.15		
N. acute medication doses/months		6.78	6.30		
Medication types (%)	Triptans	16.67%			
	Paracetamol	8.33%			
	NSAIDs	75%			
MIDAS		25.48	23.85		
HIT-6		62.15	7.23		
ASC 12		4.21	3.54		
Gait speed (m/s)		1.16	0.17	1.21	0.15
Stance phase (% gait cycle)		59.62	1.61	58.79	3.67
Swing phase (% gait cycle)		40.80	2.94	40.62	1.45
Single support (% stance phase)		40.48	1.54	40.33	1.55
Double support (% stance phase)		9.62	1.60	9.39	1.44
Stride length (m)		1.27	0.15	1.34	0.16
Cadence (steps/minute)		108.86	5.97	111.27	8.64
Pelvic tilt (°)		3.88	1.19	4.91	1.48
Pelvic obliquity (°)		8.63	3.27	10.00	3.02
Pelvic rotation (°)		7.60	2.45	10.54	3.60
sLLE_AP_		1.08	0.35	0.88	0.23
sLLE_ML_		1.17	0.27	0.96	0.18
sLLE_V_		1.16	0.31	0.98	0.19

**Table 2 sensors-24-07627-t002:** ROC curve and cutoff analyses. sLLE, short-term maximum Lyapunov’s exponent; AP, antero-posterior direction of the lower trunk acceleration signal; ML, medio-lateral direction; V, vertical direction; AUC, area under the ROC curve; OCP, optimal cutoff point; LR+, positive likelihood ratio; LR−, negative likelihood ratio; PTP+, positive post-test probability; PTP−, negative post-test probability; and PTPadj, post-test probabilities adjusted on the 12% prevalence of episodic migraine in the general population as the pre-test probability.

	AUC (95% CI)	OCP	LR+	LR−	PTP + adj	PTP − adj
sLLE_AP_	0.68 (0.55; 0.78)	≥1.10	3.36	0.61	31%	8%
sLLE_ML_	0.73 (0.60; 0.82)	≥1.18	20.32	0.47	73%	6%
sLLE_V_	0.65 (0.51; 0.76)	≥1.09	2.12	0.59	22%	7%
Pelvic tilt	0.67 (0.52; 0.77)	≤3.70	2.45	0.69	25%	9%
Pelvic rotation	0.71 (0.57; 0.82)	≤11.50	17.07	0.63	70%	8%
Stride length	0.66 (0.50; 0.77)	≤1.29	2.30	0.58	24%	7%

## Data Availability

The data that support the findings of this study are available from the corresponding author upon reasonable request.

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
