# Peer review of "Local Dynamic Stability of Trunk During Gait Can Detect Dynamic Imbalance in Subjects with Episodic Migraine"

_sensors, 2024, doi:10.3390/s24237627_

Round 1
Reviewer 1 Report
Comments and Suggestions for Authors
The authors present a valuable prospective study exploring advanced nonlinear metrics in gait analysis among migraineurs. The sample size is sufficient, the measurement and signal analysis methods are robust, and the statistical analysis is comprehensive. However, I have a few minor concerns:
-
This study was conducted during the interictal phase, and this should be clearly stated in the abstract.
-
Could you clarify the rationale behind conducting such an extensive ophthalmological examination on participants and the exclusion of individuals with glaucoma?
-
Which software did you use to create the raincloud plots (Python, JASP, or MATLAB)? Please specify this in the figure legend or in the Methods section.
-
It would be helpful to clarify the sensitivity of the sLLE metric compared to traditional gait parameters for distinguishing patients from controls. From Table 2, the sLLE metric does not appear to contribute substantially to diagnostic accuracy. However, as it reflects a distinct dynamic aspect, it may offer relevant insights in pathophysiological terms.
-
Migraineurs are known to experience marked instability and cerebellar-like disequilibrium during migraine attacks. It would strengthen the Discussion section to incorporate relevant literature on ictal posturography findings that document such instability.
Author Response
Comment 1: This study was conducted during the interictal phase, and this should be clearly stated in the abstract.
Response 1: Added.
Comment 2: Could you clarify the rationale behind conducting such an extensive ophthalmological examination on participants and the exclusion of individuals with glaucoma?
Response 2: We thank the Reviewer for giving us the opportunity to clarify this point. In collaboration with the Bietti Foundation for Ophthalmology, we performed a series of neuro-ophthalmologic examinations in order to rule out any visual factors that might affect proper walking of the patient. This was mainly because in the process of reviewing 2 previous articles using the same method but in different patients, we had been asked precisely whether we were sure that the patients did not have ophthalmologic impairments, including glaucoma, capable of affecting gait examination.
Comment 3: Which software did you use to create the raincloud plots (Python, JASP, or MATLAB)? Please specify this in the figure legend or in the Methods section.
Response 3: The raincloud plots have been created using JASP software, version 0.17.02. We added this information to the Figure legend, accordingly.
Comment 4: It would be helpful to clarify the sensitivity of the sLLE metric compared to traditional gait parameters for distinguishing patients from controls. From Table 2, the sLLE metric does not appear to contribute substantially to diagnostic accuracy. However, as it reflects a distinct dynamic aspect, it may offer relevant insights in pathophysiological terms.
Response 4: In this revised version of the Discussion section, we further discussed the point arouse by the Reviewer. From the results in Table 2, only pelvic rotation and sLLE in the medio – lateral direction reached adequate probability to correctly identify patients. In this way, as previously mentioned in the previous version of the manuscript, one possible explanation of the gait behaviour of subjects with MO may be represented by the stiffening of axial movements, in the context of cautious gait pattern due to perceived pain and instability due to multisensory integration impairment, thus increasing local instability of the trunk acceleration patterns. Since the other traditional gait parameters did not show to characterize this mechanism, the sLLEML may be considered a useful tool for identifying subjects with MO who experience dynamic instability during gait.
Specifically, we wrote, in the revised “Discussion” section:
“The other spatiotemporal gait parameters showed no significant differences when compared to healthy subjects, except for a significant reduction in stride length (Fig. 2), which did not show to characterize the gait abnormalities of participants with MO with adequate probability (Table 2). In this way, given sLLEML's ability to reflect the inability to cope with small perturbations, it may be considered a useful tool for iden-tifying subjects with MO who experience dynamic instability during gait.”
Comment 5: Migraineurs are known to experience marked instability and cerebellar-like disequilibrium during migraine attacks. It would strengthen the Discussion section to incorporate relevant literature on ictal posturography findings that document such instability.
Response 5: We thank the Reviewer for that suggestion. Now, we added the following sentence at the end of the Conclusions: “In light of the previous evidence of increased postural instability during the migraine attack compared with the interictal phase (Anagnostou et al., Eur J Neurol 2019), it is of interest to test whether the abnormalities noted here during the interictal period become more manifest during the attack and in proportion to its severity.” We added one more reference to the list:
Anagnostou E, Gerakoulis S, Voskou P, Kararizou E. Postural instability during attacks of migraine without aura. Eur J Neurol. 2019 Feb;26(2):319-e21. doi: 10.1111/ene.13815. Epub 2018 Oct 26.
Reviewer 2 Report
Comments and Suggestions for Authors
The present manuscript includes an original research study relevant to the scientific area of postural imbalance in migraine, a field for which there are limited studies, especially regarding the use of measures such as the largest Lyapunov’s exponent. It is well-structured. The methodology is well-organized and appropriately designed so as to test the hypothesis. Nevertheless, there are not included individuals with migraine older than 50 y.o., and accordingly older healthy subjects
The conclusions from the study’ results are consistent with the evidence and the initial hypothesis posed by the authors, since they answer the question whether there are differences between healthy controls and individuals with migraine without aura and an episodic pattern, by using the sLLE measure. Furthermore, they provide clinicians with a measure which has been rarely used previously for the postural instability in migraine, thus constituting the present study clinically applicable and innovative.
Some minor recommendations:
- Explain better the "Methods" Section, i.e. "sLLEML" and the "Migraine Disability Assessment score" referred in the "Results" are not properly described in the Methods.
- Provide with a brief explanation of the "somatosensory orientation test" and its correlation with migraine in the "Introduction" section.
-The references used are appropriate. Some references which could have been also used are the following:
The maximum Lyapunov exponent during walking and running: reliability assessment of different marker-sets. Ekizos A, Santuz A, Schroll A, Arampatzis A. Frontiers in Physiology. 2018. doi: 10.3389/fphys.2018.01101.
Postural instability during attacks of migraine without aura. Anagnostou E, Gerakoulis S, Voskou P, Kararizou E.Eur J Neurol. 2019 Feb;26(2):319-e21. doi: 10.1111/ene.13815.
Body balance at static posturography in vestibular migraine. Gorski LP, Silva AMD, Cusin FS, Cesaroni S, Ganança MM, Caovilla HH. Braz J. Otorhinolaryngol. 2019 Mar-Apr;85(2):183-192. doi: 10.1016/j.bjorl.2017.12.001.
Postural Instability Induced by Visual Motion Stimuli in Patients With Vestibular Migraine. Lim YH, Kim JS, Lee HW, Kim SH. Front Neurol. 2018 Jun 7;9:433. doi: 10.3389/fneur.2018.00433.
-The tables/figures show properly the data included in the manuscript are comprehensive.
Author Response
Comment 1: Explain better the "Methods" Section, i.e. "sLLEML" and the "Migraine Disability Assessment score" referred in the "Results" are not properly described in the Methods.
Response 1: We better described the sLLE and the MIDAS scale in the revised “methods” section, accordingly, and added the corresponding references.
Specifically, we modified the following sections:
“The short-term maximum finite-time Lyapunov’s exponents (sLLE) is a nonlinear metric that represents the stability of a dynamic system as the average logarithmic rate of divergence between the system's trajectory and its closest neighboring trajectory. Convergent trajectories indicate local dynamic stability, while divergent trajectories indicate local dynamic instability. As a result, when applied to gait data, it represents a measure of the ability to cope with internal perturbations during gait, thus reflecting local dynamic instability [35,36]. It was calculated based on the acceleration patterns for antero -posterior (sLLEAP), medio-lateral (sLLEML), and vertical (sLLEV) directions according to the Rosenstein’s algorithm using the Lyaprosen toolbox for nonlinear time series analysis in MATLAB environment (MATLAB 7.4.0, MathWorks, Natick, MA, USA). Twenty consecutive strides were considered for the calculations, and the acceleration signals were time-normalized to obtain 100 datapoints per stride [37]. The embedding dimension was calculated using the false nearest neighbor method, and time delay was calculated as the first minimum of the average mutual information (AMI) function [37]. Consequently, in this study an embedding dimension = 5 and time delay = 10 were used for multidimensional state - space reconstruction from the recorded one-dimensional time-series data by juxtaposing the original data and delayed copies (Fig.1). Higher sLLE values reflect higher local dynamic instability.”
As well as, regarding the MIDAS score, we added the following:
“Subjects were administered the migraine disability assessment (MIDAS), the most commonly used questionnaire to assess 3-month migraine-related disability, and the headache impact test (HIT-6) scales to assess the impact of migraine on functioning of subjects over one month, the 12-item Allodynia Symptom Checklist (ASC-12), and the numeric pain rating scale (NPRS) to record the perceived pain during migraine attacks [28-30].”
Comment 2: Provide with a brief explanation of the "somatosensory orientation test" and its correlation with migraine in the "Introduction" section.
Response 2: Added as follows: “Several studies performed in migraine the sensory organization test, which is performed through a dynamic posturography system designed to assess quantitatively an individual’s ability to use visual, proprioceptive and vestibular cues to maintain postural stability while standing. With this methodology, differences in postural balance between the subdiagnosis of migraine have been described, with greater instability and falls risk in subjects with vestibular migraine, chronic migraine, and migraine with aura [10–12], whereas subjects with episodic migraine showed less pronounced balance alterations, compared with healthy controls and subjects with chronic migraine [13,14]. However, regardless of the presence of chronic migraine and aura, subjects with migraine may experience abnormalities during mobility tasks and walking, such as reduction of gait speed and increased step width [15,16], which are common signs of dynamic imbalance [17,18].”
Comment 3: The references used are appropriate. Some references which could have been also used are the following:
The maximum Lyapunov exponent during walking and running: reliability assessment of different marker-sets. Ekizos A, Santuz A, Schroll A, Arampatzis A. Frontiers in Physiology. 2018. doi: 10.3389/fphys.2018.01101.
Postural instability during attacks of migraine without aura. Anagnostou E, Gerakoulis S, Voskou P, Kararizou E.Eur J Neurol. 2019 Feb;26(2):319-e21. doi: 10.1111/ene.13815.
Body balance at static posturography in vestibular migraine. Gorski LP, Silva AMD, Cusin FS, Cesaroni S, Ganança MM, Caovilla HH. Braz J. Otorhinolaryngol. 2019 Mar-Apr;85(2):183-192. doi: 10.1016/j.bjorl.2017.12.001.
Postural Instability Induced by Visual Motion Stimuli in Patients With Vestibular Migraine. Lim YH, Kim JS, Lee HW, Kim SH. Front Neurol. 2018 Jun 7;9:433. doi: 10.3389/fneur.2018.00433.
Response 3: We cited and added to the list the pinpointed references in the revised “Methods” and “Discussion” section. Thank you.
Comment 4: The tables/figures show properly the data included in the manuscript are comprehensive.
Response: Thank you.